# Adolescent pregnancy in Mongolia: Evidence from Mongolia Social Indicator Sample Survey 2013–2018

Khulan Bayaraa[1,2]*, Kingsley Agho[3,4,5], Blessing Akombi-Inyang[1]

1 School of Population Health, Faculty of Medicine, University of New South Wales, Sydney, New South Wales (NSW), Australia, 2 Department of Obstetrics and Gynaecology, Intermed Hospital, Ulaanbaatar, Mongolia, 3 School of Health Sciences, Western Sydney University, Penrith, NSW, Australia, 4 Translational Health Research Institute, Western Sydney University, Penrith, NSW, Australia, 5 African Vision Research Institute, University of KwaZulu-Natal, Durban, South Africa

* khulanbayaraa11@gmail.com

**Data Availability Statement:** The authors confirm that all data underlying the findings described in the manuscript are fully available without restriction. This study is based on secondary data

## Abstract

### Introduction

Adolescent pregnancy is a major public health and social concern which pose enormous pregnancy and delivery-related risks for both the mother and their neonate. This study aims to estimate adolescent pregnancy and determine the factors associated with adolescent pregnancy in Mongolia.

### Methods

This study pooled data from 2013 and 2018 Mongolia, Social Indicator Sample Surveys (MSISS). A total of 2808 adolescent girls aged 15–19 years with socio-demographic information were included in this study. Adolescent pregnancy is defined as pregnancy in a female 19 years of age or younger. Multivariable logistic regression analysis was performed to determine factors associated with adolescent pregnancy in Mongolia.

### Results

Adolescent pregnancy was estimated at 57.62 [95% Confidence Interval (CI): 44.41, 70.84] per 1000 adolescent girls aged 15–19 years. Multivariable analyses reported higher adolescent pregnancy in the countryside [Adjusted Odds Ratios (AOR) = 2.07 (95%CI: 1.08, 3.96)], with increasing age [AOR = 11.50 (95%CI: 6.64, 19.92)], among adolescent girls who used contraception methods [AOR = 10.80 (95%CI: 6.34, 18.40)], among adolescent girls from the poorest households [AOR = 3.32 (95%CI: 1.39, 7.93)], and among adolescent girls who drank alcohol [AOR = 2.10 (95%CI: 1.22, 3.62)].

### Discussion

Identifying the factors associated with adolescent pregnancy is crucial in reducing adolescent pregnancy and improving the sexual and reproductive health as well as the social and

analysis. The dataset used for this study is freely accessible online via the following link: http://web.nso.mn/nada/index.php/catalog/SISS.

**Funding:** The authors received no specific funding for this work.

**Competing interests:** The authors have declared that no competing interests exist.

economic well-being of adolescents thus setting Mongolia on the path to achieving the Sustainable Development Goals (SDGs) 3 by 2030.

## Introduction

Adolescent pregnancy is a major health and social concern which pose enormous pregnancy, birth, and development risks to both the mother and the baby [1]. It is defined as pregnancy in a female 19 years of age or younger [1]. Adolescence marks the transition from childhood to adulthood and could be classified into early adolescence (10–14 years) and late adolescence (15–19 years) [1]. During this transition period, physiological, mental and behavioural changes caused by hormone changes and neurodevelopment occur which could be influenced by various factors, some of which are related to culture and society [2, 3]. These changes may predispose the adolescent girl child to pregnancy; thus, increasing the risk of pregnancy-related health, social and economic consequences. The negative social and economic impact of adolescent pregnancy is great and long-lasting with education and employment opportunities most often compromised [4], resulting in low socioeconomic status in adulthood. In addition, adolescent pregnancy has been linked to rejection from family and community, domestic violence, bullying, stigma, suicide, and homicide [1].

It is estimated that approximately 21 million girls aged between 15 and 19 years become pregnant in low- and middle-income countries (LMICs), of which around 12 million live births and 5.6 million abortions are reported each year [5]. Globally, the adolescent birth rate has declined from 86 per 1000 female adolescents aged 15–19 in 1960 to 42 in 2018 [6]. However, this decline is not evenly distributed but vary between countries reporting its highest in LMICs and its lowest in high income countries (HICs) [7, 8]. Between 2008–2011, the lowest adolescent pregnancy rate was reported in Switzerland at 8 per 1000 adolescent girls aged 15–19 years while the highest was in Burkina Faso at 187 per 1000 adolescent girls aged 15–19 years [9]. Furthermore, South-East Asia reported the highest adolescent fertility rate of 33.6 per 1000 adolescent girls with country estimates ranging from 0.3 in the Democratic People's Republic of Korea to 83 in Bangladesh within the region [10].

In Mongolia, the adolescent birth rate increased from 18.7 per 1000 adolescent girls in 2007 to 33.1 in 2016 as reported by the Ministry of Health Mongolia [11]. This birth rate was estimated higher in the 2013 and 2018 Mongolia Social Indicator Sample Surveys (MSISS) report, at 40 and 43 per 1000 adolescent girls respectively [12, 13]. This difference in estimates could be due to methodological differences in data collection. However, no recent estimate for adolescent pregnancy has been provided either by the Ministry of Health Mongolia or MSISS. Hence, our study will estimate adolescent pregnancy using the 2013 and 2018 MSISS.

Adolescent pregnancy is one of the leading causes of death among young women specifically in LMICs [14]. An increased risk of pregnancy complications including preeclampsia, preterm birth, postpartum haemorrhage and infections during pregnancy, childbirth and post-delivery that lead to maternal mortality and morbidity are reported among the youngest pregnant adolescents [10, 15]. Studies have shown that severe prenatal and neonatal conditions such as stillbirth and neonatal death related to low birth weight, preterm birth, and other complications around labour are higher among babies born to teenage mothers compared to their counterparts aged between 20–24 years old [10]. In addition, unsafe abortion is another negative consequence of adolescent pregnancy that is often underreported.

Common predisposing factors of adolescent pregnancy are low educational attainment, low socioeconomic status, child marriage or early union, risky health behaviours and lack of

reproductive knowledge. Moreover, rural residency, employment status, non-supportive family and school environment, peer pressure and social, cultural, and traditional norms are region- and country- specific drivers of adolescent pregnancy as well [2, 3, 16–19]. Hence, this study will estimate adolescent pregnancy and determine the factors associated with adolescent pregnancy in Mongolia using the 2013 and 2018 MSISS. Findings from this study would be helpful to policymakers and public health researchers in formulating effective interventions to reduce adolescent pregnancy by strategically targeting its predisposing factors.

## Methods

### Data source

This study used pooled data used from the 2013 and 2018 MSISS. The surveys were conducted as part of the global MICS programme and implemented by the National Statistics Office of Mongolia (NSO) in conjunction with the Mongolian Government, United Nations Children's Fund (UNICEF) and United Nations Population Fund (UNFPA). MSISS collected data to compile indicators in health, education, development, protection, well-being and rights of children and women. In addition, data on reproductive health, family planning, knowledge and attitude towards HIV/AIDS, and sexual behaviour of Mongolian men and women were collected. Structured survey questionnaires were used to collect relevant socio-demographic, individual and reproductive and sexual health information. The surveys reported a response rate of 98.3% and 98.5% in 2013 and 2018, respectively. Full details of sampling techniques and other relevant information have been published elsewhere [12, 13].

A total of 15500 households were selected for the survey in 2013 and 14500 households in 2018 using a two-stage, stratified cluster sampling technique. Of which 12830 women aged 15–49 years were interviewed in 2013 and 10794 in 2018. For this study, we combined the 2013 and 2018 data sets by making sure the variable names in two the data sets were coded and labelled the same and 'append' command in STATA was used to combine the two cross-sectional data sets to have a enough power to detect any statistical differences and we then restricted our analyses to the 2808 adolescent women aged 15–19 years (1589 in 2013 and 1219 in 2018) with valid information (Fig 1).

### Dependent variable

The outcome variable of this study was adolescent pregnancy, defined as women aged 15–19 years who reported ever being pregnant previously or at the time of the interview. Information on pregnancy outcomes and current pregnancy status obtained from the surveys were used to construct the outcome variable.

### Independent variables

Independent variables were selected from the 2013 and 2018 MSISS based on the World Health Organization (WHO) conceptual frameworks for action on the social determinants of health [20]. The explanatory factors were classified into four main categories: community-level factors, socio-demographic factors, individual-level factors, and media factors as shown in Fig 2.

Community-level factors included type of residence and geographical zones. Socio-demographic factors included age, education level of adolescent girls, marital status, ethnicity, religion, and wealth index. The household wealth index serves as an indicator of wealth consistent with expenditure and income measures and was constructed based on principle components analysis (PCA) using household assets. The wealth index was categorised into five wealth quintiles including poorest, poor, middle, rich, and richest equally [12, 13, 21]. Individual-level

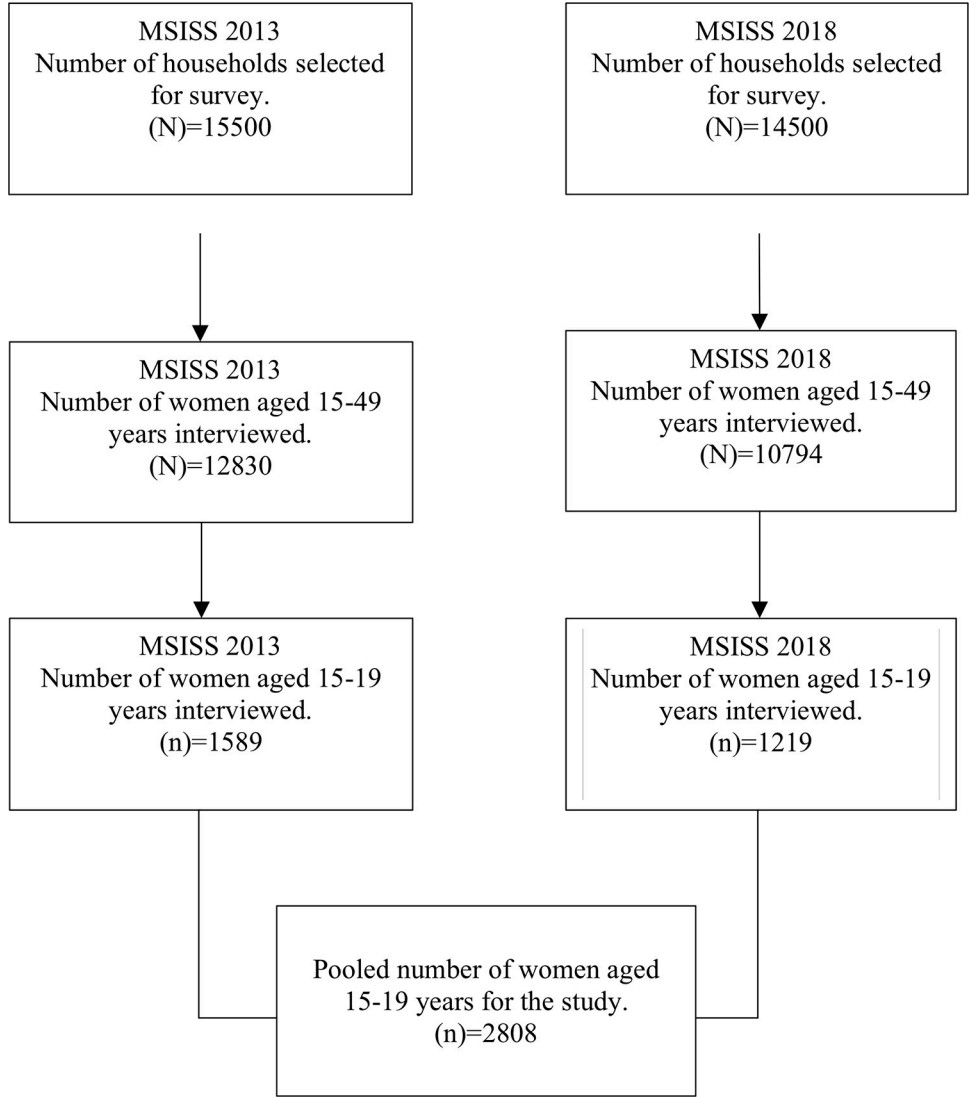

**Fig 1. Flow chart for the selection of the study population.**

factors were experience of smoking tobacco and drinking alcohol. Media factors included the frequency of reading the newspaper, listening to the radio, watching TV, the experience of using a computer or tablet, access to the internet and possession of a mobile phone.

## Statistical analysis

To determine the factors associated with adolescent pregnancy, the dependent variable was expressed as a dichotomous variable i.e., category 1 [Adolescent pregnancy] and category 0 [No adolescent pregnancy].

Stata version 16.0 was used to perform the data analyses (StataCorp, College Station, TX, USA). Survey tabulation using the Taylor series linearization method was used to estimate adolescent pregnancy and the confidence intervals (CIs) of key exploratory factors among women aged 15–19 years this was followed with simple and multiple logistic regression analyses.

**Community level Factors**

-Type of residency

-Geographical Zone

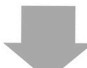

**Socio-demographic factors**

-Age

-Education level

-Marital Status

-Religion of household head

-Ethnicity of household head

-Wealth index

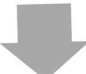

**Individual Level factors**

-Experience of smoking

-Experience of drinking alcohol

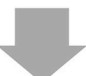

**Media Factors**

-Frequency of reading the newspaper

-Frequency of listening to the radio

-Frequency of watching TV

-Experience of using a computer/tablet

-Experience of using the internet

-Possesion of a mobile phone

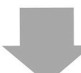

**Adolescent pregnancy**

**OR**

**No adolescent pregnancy**

**Fig 2. A modified conceptual framework for factors associated with adolescent pregnancy in Mongolia.**

Univariable and bivariate logistic regression model analyses were used to determine the unadjusted estimate between the exploratory factors and adolescent pregnancy. Thereafter, multiple logistic regression analyses were conducted using a four-stage conceptual modelling technique as described in Fig 2. Only those exploratory factors with a p-value less than 0.20 obtained from univariable and bivariate analyses were entered into the multivariable logistic regression analyses. In the first stage, community-level factors were entered into the baseline model to assess their relationship with the study outcome variable. A manual stepwise backward elimination method was conducted, and a factor significantly associated with the study outcome was retained. In the second stage, socio-demographic factors were added to the significant factor from the first model and a similar manual stepwise backward elimination procedure was performed. This sequential removing technique was repeatedly used to include individual-level and media factors in the third and fourth stages, respectively. The factors with p-values less than 0.05 were retained as the factors associated with adolescent pregnancy. To avoid any statistical bias, potential factors with p-value less than 0.20 obtained from bivariate analysis were entered into multivariable analysis. The adjusted variables were assessed by calculating the odds ratios with 95%C and those with p<0.05 remained in the final model.

## Ethics

This study is based on secondary data analysis. The first author applied for and obtained permission from Mongolian National Statistical Office (NSO) to download and use the data for this study. Hence, no ethics approval was required.

## Results

### Characteristics of the sample

A total sample of 2808 adolescent girls aged 15–19 years were included in the study. Of those, 71.6% lived in urban areas while 28.4% lived in the countryside. 99.7% were currently unmarried. About 41.4% had no religion and the majority (78.2%) were Khalkha ethnic group while approximately 16.5% were in the poorest and 20.7% were in the richest wealth index quintile. 80.9% never smoked and 68.6% never drank alcohol (Table 1).

Furthermore, 14.8% of this study sample, reported having sex. Of those, 8.8% were younger than 15 years old and 39.1% experienced a pregnancy (Table 2). Separate descriptive analysis was also conducted for weighted and unweighted data for both 2013 and 2018 surveys. In both 2013 and 2018 surveys, the frequency distribution was similar and the percentage of participants in urban residence were about one-third of those participants in Countryside (see, Tables A & B in S1 Text for details).

Fig 3 shows adolescent pregnancy in Mongolia. Adolescent pregnancy was 63.69 [95%CI: 50.85, 76.53] and 50.13 [95%CI: 34.65, 65.60] per 1000 adolescent girls aged 15–19 years in 2013 and 2018, respectively. As there was no statistically significant difference between MSISS adolescent pregnancy rate in 2013 and 2018, the pooled adolescent pregnancy rate for 2013 and 2018 was estimated at 57.62 [95%CI: 44.41, 70.84] per 1000 adolescent girls aged 15–19 years.

### Factors associated with Adolescent pregnancy

Adolescent pregnancy was higher in the countryside compared to urban areas. Adolescent pregnancy increased with age. Girls aged 18–19 years were more likely to become pregnant compared to younger girls aged 15–17 years. Adolescent girls who used contraceptive methods were more prone to pregnancy compared to girls who did not use any form of contraception.

**Table 1. Characteristics of the sample, prevalence and 95% confidence intervals.** Mongolia SISS 2013 and 2018 (N* = 2808).

| Study variables | Unweighted Number N (%) | Weighted Number n (%) | Prevalence (95%CI) |
|---|---|---|---|
| Year of survey (N = 2808) | | | |
| 2013 | 1589 (56.6) | 1599 (56. 9) | 6.4 (5.2, 7.8) |
| 2018 | 1219 (43.4) | 1209 (43.1) | 5.0 (3.7, 6.8) |
| *Community level factors* | | | |
| Type of residence (N = 2808) | | | |
| Urban | 1769 (63.0) | 2010 (71.6) | 5.2 (4.1, 6.5) |
| Countryside | 1039 (37.0) | 798 (28.4) | 7.3 (5.6, 9.3) |
| Geographical zones (N = 2808) | | | |
| Western | 589 (43.7) | 380 (13.5) | 1.8 (0.8, 3.9) |
| Khangai | 548 (38.8) | 489 (17.4) | 6.3 (4.5, 8.9) |
| Central | 354 (25.0) | 338 (12.0) | 9.6 (6.7, 13.5) |
| Eastern | 333 (23.9) | 178 (6.3) | 11 (7.9, 15.1) |
| Ulaanbaatar/Capital city | 984 (68.4) | 1423 (50.7) | 1.8 (0.8, 3.9) |
| *Socio-demographic factors* | | | |
| Age (N = 2808) | | | |
| 15–17 | 2040 (72.7) | 1934 (68.9) | 1.2 (0.7, 1.8) |
| 18–19 | 768 (27.4) | 874 (31.1) | 15.9 (13.2, 19.2) |
| Education level (N = 2807) | | | |
| No Schooling | 44 (3.0) | 42 (1.5) | 14.6 (6.5, 29.7) |
| Primary/basic | 1808 (130.4) | 1630 (58.1) | 1.2 (0.8, 1.9) |
| Secondary or more | 955 (66.6) | 1135 (40.4) | 12.1 (9.9, 14.6) |
| Marital Status (N = 2808) | | | |
| Currently unmarried | 2798 (99.7) | 2800 (99.7) | 5.6 (4.7, 6.6) |
| Formerly married | 10 (0.4) | 8.0 (0.3) | 78.7 (40.2, 95.3) |
| Religion (N = 2801) | | | |
| No religion | 1127 (40.2) | 1159 (41.4) | 7.4 (5.9, 9.2) |
| Buddha | 1352 (48.3) | 1394 (49.8) | 4.5 (3.3, 5.9) |
| Other | 322 (11.5) | 246 (8.8) | 5.6 (3.0, 10.0) |
| Ethnicity (N = 2803) | | | |
| Khalkha | 2080 (74.3) | 2191(78.2) | 6.1 (5.0, 7.4) |
| Kazakh | 232 (8.3) | 129 (4.61) | 0.9 (0.2, 5.2) |
| Other | 721 (17.5) | 482 (17.2) | 5.4 (3.6, 8.1) |
| Wealth index (N = 2808) | | | |
| Poorest | 630 (22.4) | 464 (16.5) | 8.5 (6.3, 11.3) |
| Poor | 665 (23.7) | 563 (20.1) | 6.9 (4.9, 9.5) |
| Middle | 568 (20.2) | 561 (20.0) | 5.9 (4.0, 8.8) |
| Rich | 525 (18.7) | 639 (22.8) | 3.9 (2.4, 6.4) |
| Richest | 420 (14.9) | 581 (20.7) | 4.4 (2.7, 7.1) |
| *Individuals level factors* | | | |
| Ever smoked tobacco (N = 2804) | | | |
| No | 2358 (84.1) | 2271 (80.9) | 4.9 (3.9, 6.0) |
| Yes | 446 (15.9) | 533 (19.1) | 9.7 (7.1, 13.1) |
| Ever drunk alcohol (N = 2801) | | | |
| No | 2032 (72.6) | 1922 (68.6) | 3.5 (2.7, 4.5) |
| Yes | 769 (27.5) | 879 (31.4) | 10.9 (8.7, 13.6) |
| Combined tobacco and alcohol (N = 2797) | | | |
| None | 1889 (67.5) | 1774 (63.4) | 3.23 (2.5, 4.3) |

*(Continued)*

**Table 1.** (Continued)

| Study variables | Unweighted Number N (%) | Weighted Number n (%) | Prevalence (95%CI) |
|---|---|---|---|
| Only tobacco | 139 (5.0) | 147 (5.3) | 6.4 (3.2, 12.4) |
| Only alcohol | 462 (16.5) | 491 (17.5) | 10.8 (8.0, 14.5) |
| Both | 307 (11.0) | 388 (13.9) | 10.9 (7.6, 15.4) |
| *Media factors* | | | |
| Reading the newspaper (N = 2795) | | | |
| Not at all | 1025 (36.7) | 1051 (37.6) | 5.9 (4.4, 7.9) |
| Yes | 1768 (63.3) | 1744 (62.4) | 5.7 (4.6, 7.0) |
| Listening to the radio (N = 2808) | | | |
| Not at all | 2153 (76.7) | 2146 (76.4) | 5.9 (4.8, 7.2) |
| Yes | 655 (23.3) | 662 (23.6) | 5.5 (3.9, 7.7) |
| Watching TV (N = 2805) | | | |
| Not at all | 158 (5.6) | 134 (4.8) | 6.8 (6.8, 13.1) |
| Yes | 2647 (94.4) | 2672 (95.2) | 5.7 (4.8, 6.9) |
| Ever used a computer (N = 2805) | | | |
| No | 496 (17.7) | 399 (14.2) | 8.3 (5.9, 11.5) |
| Yes | 2309 (82.3) | 2405 (85.8) | 5.4 (4.4, 6.5) |
| Ever used the internet (N = 2808) | | | |
| No | 401 (14.3) | 353 (12.6) | 8.8 (6.3, 12.2) |
| Yes | 2407 (85.7) | 2455 (87.4) | 5.4 (4.4, 6.6) |
| Has a mobile phone (N = 2808) | | | |
| No | 148 (5.3) | 120 (4.3) | 5.5 (2.3, 12.5) |
| Yes | 2660 (94.7) | 2688 (95.7) | 5.8 (4.9, 6.9) |

Adolescent girls from poor households were highly likely to become pregnant compared to those from more affluent households. In addition, adolescent girls who drank alcohol were more susceptible to adolescent pregnancy (Table 3). Conducting a separate yearly analysis, our result indicated that adolescent girls aged 18–19 years, adolescent girls who used contraceptive methods and who smoked tobacco and drank alcohol were significant in both the 2013 and 2018 surveys, respectively (see, Tables A & B in S2 Text for details). Additionally, in a 2013 survey, it was found that adolescent pregnancies were 88% more likely to occur among Buddhists than those with no religion [adjusted OR = 1.88, 95%CI: 1.10, 3.20], see Table A in S2 Text for details).

## Discussion

This study indicates that residence in the countryside, increasing age, using contraception methods, being from a poor household and drinking alcohol were the main factors associated with adolescent pregnancy in Mongolia.

The likelihood of becoming pregnant during adolescence was higher among countryside residents compared to urban residents. A plausible explanation could be the limited access to education due to the presence of fewer educational facilities in the countryside. With fewer schools, only a restricted number of adolescents can obtain proper education. The interplay between place of residence, educational attainment, and adolescent pregnancy is quite complex. Applying the principle of intersectionality [22], it could be inferred that adolescent girls, with no/low level of educational attainment, and residing in the countryside are more susceptible to pregnancy compared to their counterparts in urban areas. In addition, most countryside have fewer job opportunities [23], which impacts the economic stability of most families,

**Table 2. Sexual and reproductive health characteristics among adolescent girls aged 15–19 years in Mongolia, SISS 2013 and 2018 (N = 415).**

| Characteristics | Number (n) | Percentage (%) |
|---|---|---|
| Ever had sex (N = 2808) | | |
| No | 2393 | 85.2 |
| Yes | 415 | 14.8 |
| *Age at first sex (N = 415) | | |
| 15–17 | 247 | 59.5 |
| 18–19 | 168 | 40.5 |
| *Ever experienced a pregnancy (N = 415) | | |
| No | 253 | 60.9 |
| Yes | 162 | 39.1 |
| *Ever give a birth (N = 415) | | |
| No | 317 | 76.2 |
| Yes | 98 | 23.8 |
| *Ever had abortion, miscarriage, and stillbirth (N = 415) | | |
| No | 385 | 92.8 |
| Yes | 30 | 7.2 |
| *Currently pregnant (N = 415) | | |
| No | 373 | 90.0 |
| Yes | 42 | 10.0 |
| *Ever used any contraceptive method (N = 415) | | |
| No | 246 | 59.3 |
| Yes | 169 | 40.7 |

*restricted only to adolescent girls who have ever had sex.

leading to parents marrying off their daughters at an earlier age for economic security [24] thus increasing the likelihood of adolescent pregnancy [25]. This study reported that adolescent pregnancy was associated with increasing adolescent age with girls aged 18–19 years being more susceptible. This finding is consistent with previous studies in LMICs [2] which reported an association between increasing adolescent age and pregnancy. The increase in adolescent pregnancy with age could be due to the fact that in Mongolia, under Family Law 1999, the minimum legal age for marriage is 18 years [26], hence, pregnancy after 18 is socially accepted which gives rise to an increase in sexual activity among adolescents thus leading to

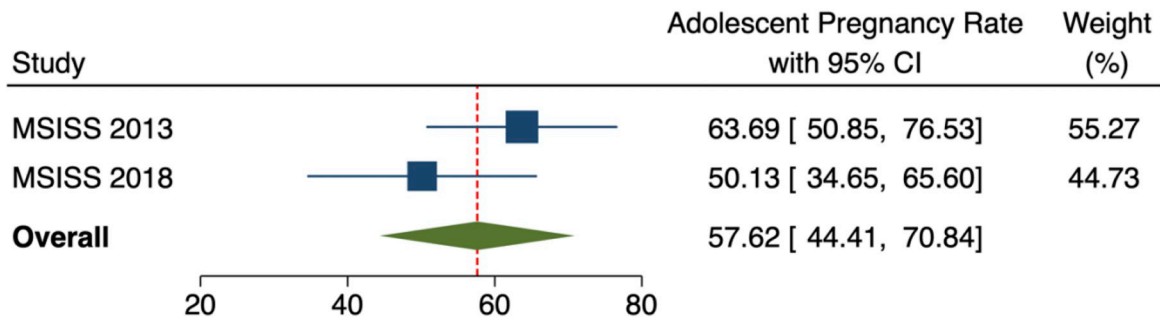

**Fig 3. Adolescent pregnancy in Mongolia, 2013–2018.**

**Table 3.  Unadjusted and adjusted odds ratios (OR) with 95%CI for factors associated with adolescent pregnancy in Mongolia, MSISS 2013 and 2018 (N = 2794).**

| Study variable | Unadjusted Odd Ratio (UOR) [95%Cl] | P-value | Adjusted Odd Ratio (AOR) [95%Cl] | P-value |
|---|---|---|---|---|
| **Year of survey** *(2013, OR = 1)* | | | | |
| 2018 | 0.95 [0.55, 1.67] | 0.869 | 0.91 [0.55, 1.49] | 0.706 |
| **Type of residence** *(Urban, OR = 1)* | | | | |
| Countryside | 2.21 [1.05, 4.62] | 0.036 | 2.07 [1.08, 3.96] | 0.028 |
| **Age in categories** *(15–17 years, OR = 1)* | | | | |
| 18–19 | 6.10 [2.99, 12.43] | <0.001 | 11.50 [6.64, 19.92] | <0.001 |
| **Ever used contraceptive methods** *(No, OR = 1)* | | | | |
| Yes | 23.08 [15.15, 35.16] | <0.001 | 10.80 [6.34, 18.40] | <0.001 |
| **Household Wealth index** *(Richest, OR = 1)* | | | | |
| Poorest | 3.79 [1.57, 9.13] | 0.003 | 3.32 [1.39, 7.93] | 0.007 |
| Poor | 2.33 [1.07, 5.04] | 0.033 | 2.41 [1.08, 5.34] | 0.031 |
| Middle | 2.16 [1.03, 4.53] | 0.042 | 2.81 [1.35, 5.88] | 0.006 |
| Rich | 1.14 [0.51, 2.52] | 0.753 | 1.15 [0.51, 2.60] | 0.737 |
| **Combined tobacco and alcohol** *(None, OR = 1)* | | | | |
| Only tobacco | 1.09 [0.26, 4.58] | 0.909 | 1.43 [0.41, 4.94] | 0.571 |
| Only alcohol | 1.67 [0.91, 3.05] | 0.095 | 2.10 [1.22, 3.62] | 0.007 |
| Both | 1.21 [0.57, 2.57] | 0.619 | 1.52 [0.75, 3.09] | 0.248 |

pregnancy. In South Asia, adolescent pregnancy is known as being associated with child marriage, especially as it relates to traditional practices within the region [8, 27]. There is usually a loss of sexual autonomy with child marriage which may lead to increased sexual activity among married adolescents. This increase in sexual activity could increase the likelihood of pregnancy among adolescent girls especially where there is an absence or incorrect knowledge of contraception [1]. Given the association between child marriage/union and adolescent pregnancy, more commitment to eliminating child marriage/union is needed at the family, community and policymakers' level to reduce adolescent pregnancy and give the adolescent a future with more study and work opportunity [28].

This study also reported that adolescent pregnancy was higher among adolescent girls who used contraception methods. Many adolescents engage in sexual activity and use contraception but may not be fully informed about how to properly use their preferred contraception method. Impulsivity, lack of planning, and concurrent drug and alcohol use decrease the likelihood that adolescents will use birth control and barrier protection properly [29]. One of the most common problems of contraception use among adolescents is adherence. This may involve forgetting to take daily oral contraceptives or stopping them entirely often without substituting another form of birth control [29] which increases the risk of pregnancy.

Our study also showed that adolescent girls from poor households were more susceptible to pregnancy compared to their counterparts from richer households. This relationship between adolescent pregnancy and low economic status has been well-established in previous studies [7, 8, 30–32].

This study reported adolescent pregnancy among adolescent girls who drank alcohol. Early regular alcohol consumption is associated with early onset of sexual activity [33]. Research has shown that any amount of current drinking by adolescents is associated with being sexually active, especially binge drinking and drinking in greater quantities [34].

There is a strong association between alcohol use and risky sexual behavior, including having unprotected sex which may lead to unintended pregnancy [33]. Alcohol use at first sex is associated with lower levels of condom use at first intercourse. Furthermore, research has also

shown that alcohol consumption at an early age is strongly associated with having a higher (or multiple) number of sexual partners and sex without any contraception [33, 34]. Therefore, preventing alcohol use among adolescents may reduce risky sexual behaviours and outcomes, including adolescent pregnancy.

This study had some limitations. First, due to the cross-sectional nature of the study design, this paper is limited in its ability to establish a causal relationship between the observed associated factors and adolescent pregnancy. Second, selection bias is inherent in combining two cross-sectional data sets of this nature because, in this study, participants selected in 2013 may be different from those participants selected in 2018 due to 5 years' time lag. Third, based on data collected in MSISS, our study included adolescent girls aged 15–19 years only. As a result, this study's estimation of adolescent pregnancy might be under or overestimated. Thus, additional studies targeting adolescent girls aged 10–19 are urgently needed.

However, this study also had some strengths. First, this study has an increased statistical power as it used pooled MSISS data for the period (2013–2018), which is currently the largest sample size in Mongolia. Second, the MSISS data used for this study are nationally representative with high response rates of 98.3% and 98.5% in 2013 and 2018, respectively. Third, due to applying appropriate statistical adjustment and sampling techniques in the primary data analysis, the result of this study could be generalised to the entire Mongolian adolescent girls' subpopulation.

## Conclusion

This study shows that adolescent pregnancy results from an interaction of factors. Therefore, targeted community-based interventions which address these factors associated with adolescent pregnancy are needed. Such interventions should focus on providing comprehensive sexuality education including proper contraceptive use and the impact of alcohol use on decision-making especially among adolescent girls residing in countryside Mongolia and who are from poor households regardless of education level. In addition, a further population-based study among adolescent girls aged 10–19 years is crucial to obtain a more precise estimation and have a better understanding of the factors associated with adolescent pregnancy in Mongolia.

## Supporting information

**S1 Text. Table A:** Characteristics of the sample, prevalence and 95% confidence Intervals, Mongolia SISS 2013. **Table B:** Characteristics of the sample, prevalence and 95% Confidence Intervals (CI), Mongolia SISS 2018.
(DOCX)

**S2 Text. Table A:** Unadjusted and adjusted odds ratios (OR) with 95%CI for factors associated with adolescent pregnancy in Mongolia, MSISS 2013. **Table B:** Unadjusted and adjusted odds ratios (OR) with 95%CI for factors associated with adolescent pregnancy in Mongolia, MSISS 2018.
(DOCX)

## Acknowledgments

This study was part of the first author's Masters' degree research project with the School of Population Health, University of New South Wales (UNSW). The first author would like to acknowledge her co-supervisor Dr. Anurag Sharma from the School of Population Health, UNSW for his support throughout the project.

## Author Contributions

**Conceptualization:** Khulan Bayaraa, Blessing Akombi-Inyang.

**Data curation:** Khulan Bayaraa, Kingsley Agho.

**Formal analysis:** Khulan Bayaraa.

**Investigation:** Khulan Bayaraa, Blessing Akombi-Inyang.

**Methodology:** Khulan Bayaraa, Blessing Akombi-Inyang.

**Project administration:** Khulan Bayaraa, Blessing Akombi-Inyang.

**Resources:** Khulan Bayaraa.

**Supervision:** Blessing Akombi-Inyang.

**Validation:** Khulan Bayaraa, Kingsley Agho.

**Visualization:** Khulan Bayaraa, Blessing Akombi-Inyang.

**Writing – original draft:** Khulan Bayaraa.

**Writing – review & editing:** Khulan Bayaraa, Blessing Akombi-Inyang.

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
