## [Decision Letter · Decision Letter 0]

2 Nov 2022

PGPH-D-22-00650

Adolescent pregnancy in Mongolia: 

Evidence from Mongolia Social Indicator Sample Survey 2013-2018

Dear Dr. Bayaraa,

Thank you for submitting your manuscript to PLOS Global Public Health. After careful consideration, we feel that it has merit but does not fully meet PLOS Global Public Health’s publication criteria as it currently stands. Therefore, we invite you to submit a revised version of the manuscript that addresses the points raised during the review process.

We look forward to receiving your revised manuscript.

Kind regards,

Colleen M. Davison

Academic Editor

Journal Requirements:

2. Please amend your Data Availability Statement and indicate where the data may be found.

Reviewers' comments:

Reviewer's Responses to Questions

**Comments to the Author**

1. Does this manuscript meet PLOS Global Public Health’s publication criteria? Is the manuscript technically sound, and do the data support the conclusions? The manuscript must describe methodologically and ethically rigorous research with conclusions that are appropriately drawn based on the data presented.

Reviewer #1: Yes

Reviewer #2: Partly

2. Has the statistical analysis been performed appropriately and rigorously?

Reviewer #1: Yes

Reviewer #2: No

3. Have the authors made all data underlying the findings in their manuscript fully available (please refer to the Data Availability Statement at the start of the manuscript PDF file)?

Reviewer #1: Yes

Reviewer #2: Yes

4. Is the manuscript presented in an intelligible fashion and written in standard English?

Reviewer #1: No

Reviewer #2: Yes

5. Review Comments to the Author

Reviewer #1: Adolescent pregnancy is one of the important issues in obstetrics and gynecology. The authors have addressed about the adolescent pregnancy in Mongolia. I think the data will be interesting for both international and Mongolian readers.

There are few suggestions:

1. In table 1, the marital status of weighted prevalence of adolescent pregnancy is 2.1 and 70.8 vs 95.3 and 4.8. It is suitable to explain why the numbers are so different.

2. Age is grouped differently. Such as in table 1, 15-17,18-19 vs in table 2, it is <15, 16-17,18-19. What is the reason to change the age group?

Reviewer #2: The possibility of different interpretations arises from the fact that both individual and combined procedures will result in differing estimations. The feasibility of the unique technique should therefore be first examined by researchers, it is recommended. The pooled technique should only be used when it is logical to assume that the features and the domains of interest remain constant from cycle to cycle.

As a result, I suggest carrying out a statistical analysis for the 2013 dataset separately from the 2018 dataset. Before combining data to create pooled estimates, there are a few factors to consider. One could say that the considerations are an iterative process: analyze the rationale for why integrating is desired, ensure that integrating makes sense, integrate the data, compute estimates or conduct analysis, review the results, and determine if integrating still makes sense. Repeat. I won't be able to publish this if you don't give me adequate supporting scientific evidence.

-Please double-check and include the following before doing a pooled analysis, to be very specific. If you have previously completed this analysis, please provide a thorough explanation of the dataset merging process.

- When using pooled data to estimate the parameters in your regression analysis, keep an eye out for the impact of cycles.

- Are the 2013 and 2018 samples comparable? A five-year void exists. Because methods can change over time, the author should exercise caution.

-Which of the two sets of variables is most comparable? Are there any similarities between the two sets of estimates? Each variable should be checked independently, and the author should detail this in the paper's methods section. To determine if the two population parameters are related, the authors should do a formal hypothesis test on each set of estimations.

Please verify the following during the MLR analysis:

-According to the MLR, which has an OR of 70.1, those who live together or are married are more likely to be exposed to adolescent pregnancy. But I'm concerned that it doesn't adequately describe the whole thing. For instance, most research findings concur that married persons are more likely to have enduring relationships and more amenable to having children. Because this OR is so large and might be incorrect, please double-check your statistical analysis.

-Justify your choice to utilize the western region of Mongolia as the MLR's reference value. The western area of Mongolia has the lowest weighted prevalence of adolescent pregnancy, according to your data.

-

6. PLOS authors have the option to publish the peer review history of their article (what does this mean?). If published, this will include your full peer review and any attached files.

**Do you want your identity to be public for this peer review?** For information about this choice, including consent withdrawal, please see our Privacy Policy.

Reviewer #1: **Yes: **Batsuren Choijamts

Reviewer #2: **Yes: **MANDUKHAI GANBAT

---

## [Decision Letter · Decision Letter 1]

16 Feb 2023

PGPH-D-22-00650R1

Adolescent pregnancy in Mongolia: 

Evidence from Mongolia Social Indicator Sample Survey 2013-2018

Dear Dr. Bayaraa,

Thank you for submitting your manuscript to PLOS Global Public Health. After careful consideration, we feel that it has merit but does not fully meet PLOS Global Public Health’s publication criteria as it currently stands. Therefore, we invite you to submit a revised version of the manuscript that addresses the points raised during the review process.

Please also explain how you merged two distinct time periods in the study's procedure section and please consider the reviewer's comment about adding an additional study limitation related to this.

We look forward to receiving your revised manuscript.

Kind regards,

Colleen M. Davison

Academic Editor

Journal Requirements:

Additional Editor Comments (if provided):

Please consider the reviewer's comment about adding an additional study limitation.

Reviewers' comments:

Reviewer's Responses to Questions

**Comments to the Author**

1. If the authors have adequately addressed your comments raised in a previous round of review and you feel that this manuscript is now acceptable for publication, you may indicate that here to bypass the “Comments to the Author” section, enter your conflict of interest statement in the “Confidential to Editor” section, and submit your "Accept" recommendation.

Reviewer #2: All comments have been addressed

2. Does this manuscript meet PLOS Global Public Health’s publication criteria? Is the manuscript technically sound, and do the data support the conclusions? The manuscript must describe methodologically and ethically rigorous research with conclusions that are appropriately drawn based on the data presented.

Reviewer #2: Partly

3. Has the statistical analysis been performed appropriately and rigorously?

Reviewer #2: Yes

4. Have the authors made all data underlying the findings in their manuscript fully available (please refer to the Data Availability Statement at the start of the manuscript PDF file)?

Reviewer #2: Yes

5. Is the manuscript presented in an intelligible fashion and written in standard English?

Reviewer #2: Yes

6. Review Comments to the Author

Reviewer #2: I like that the author attempted to address the publication.

I believe the author did not explain how she or he merged two distinct time periods in the study's procedure section, which is very important to include.

Prior to publication, one thing must be carefully considered.

In the BMJ article you provide, they address a frequentist statistical hypothesis rather than a scientific hypothesis. Various types of study are more exploratory and, hence, less hypothesis-driven. Between the two time points, the prevalence declined from 6.4% to 5%, hence it can be stated that the disease is less prevalent in 2018. But without statistical analysis, it is impossible to identify whether the numbers are identical or distinct. Prevalence may be a calculation you report as a result, but you can provide a general idea of it beforehand. I hypothesize, for instance, that the prevalence is +/- 1.4% than other part of the sample or that it is strongly correlated with socioeconomic characteristics. I refer to these scientific hypotheses. Therefore, you may include it in the study's limitation section.

7. PLOS authors have the option to publish the peer review history of their article (what does this mean?). If published, this will include your full peer review and any attached files.

**Do you want your identity to be public for this peer review?** For information about this choice, including consent withdrawal, please see our Privacy Policy.

Reviewer #2: **Yes: **Mandukhai Ganbat

---

## [Decision Letter · Decision Letter 2]

22 Mar 2023

Adolescent pregnancy in Mongolia: 

Evidence from Mongolia Social Indicator Sample Survey 2013-2018

PGPH-D-22-00650R2

Dear Dr Bayaraa,

We are pleased to inform you that your manuscript 'Adolescent pregnancy in Mongolia: 

Evidence from Mongolia Social Indicator Sample Survey 2013-2018' has been provisionally accepted for publication in PLOS Global Public Health.

Best regards,

Colleen M. Davison

Academic Editor

Reviewer Comments (if any, and for reference):

Reviewer's Responses to Questions

**Comments to the Author**

1. If the authors have adequately addressed your comments raised in a previous round of review and you feel that this manuscript is now acceptable for publication, you may indicate that here to bypass the “Comments to the Author” section, enter your conflict of interest statement in the “Confidential to Editor” section, and submit your "Accept" recommendation.

Reviewer #2: All comments have been addressed

2. Does this manuscript meet PLOS Global Public Health’s publication criteria? Is the manuscript technically sound, and do the data support the conclusions? The manuscript must describe methodologically and ethically rigorous research with conclusions that are appropriately drawn based on the data presented.

Reviewer #2: Yes

3. Has the statistical analysis been performed appropriately and rigorously?

Reviewer #2: Yes

4. Have the authors made all data underlying the findings in their manuscript fully available (please refer to the Data Availability Statement at the start of the manuscript PDF file)?

Reviewer #2: Yes

5. Is the manuscript presented in an intelligible fashion and written in standard English?

Reviewer #2: Yes

6. Review Comments to the Author

Reviewer #2: Limitation section is satisfactory

Good Luck :)

7. PLOS authors have the option to publish the peer review history of their article (what does this mean?). If published, this will include your full peer review and any attached files.

**Do you want your identity to be public for this peer review?** For information about this choice, including consent withdrawal, please see our Privacy Policy.

Reviewer #2: **Yes: **Mandukhai Ganbat
